# Combination of Standard Addition and Isotope Dilution Mass Spectrometry for the Accurate Determination of Melamine and Cyanuric Acid in Infant Formula

**DOI:** 10.3390/foods13152377

**Published:** 2024-07-27

**Authors:** Vasilisa Pedan, Rudolf Koehling, Lukas Drexel, Kathrin Breitruck, Alexander Rueck, Sascha Rohn, Olaf Rienitz, Axel Pramann, Tim Seidel, Eric Allenspach, Markus Obkircher

**Affiliations:** 1Sigma-Aldrich Production GmbH (Subsidiary of Merck KGaA), Industriestrasse 25, 9471 Buchs, SG, Switzerland; vasilisa.pedan@merckgroup.com (V.P.); lukas.drexel@aon.at (L.D.); kathrin.breitruck@merckgroup.com (K.B.); alexander.rueck@merckgroup.com (A.R.); tim.seidel@merckgroup.com (T.S.); eric.allenspach@merckgroup.com (E.A.); markus.obkircher@merckgroup.com (M.O.); 2Institute of Food Technology and Food Chemistry, Department of Food Chemistry and Analysis, Technische Universität Berlin, Gustav-Meyer-Allee 25, 13355 Berlin, Germany; rohn@tu-berlin.de; 3Physikalisch-Technische Bundesanstalt, Bundesallee 100, 38116 Braunschweig, Germany; olaf.rienitz@ptb.de (O.R.); axel.pramann@ptb.de (A.P.)

**Keywords:** isotope dilution mass spectrometry, proficiency testing, infant formula, two–dimensional liquid chromatography, residues

## Abstract

In the melamine scandals of the early 2000s, different companies of the dairy industry cheated their products by applying chemical substances to feign a higher content of nitrogen. However, this had a severe toxic impact on the kidney health of consumers. As a result, tremendous effort was put into the prevention of further harm to the public. In the present study, a fast–screening method for the determination of melamine and cyanuric acid in infant formula was developed. While a 1D–LC approach is faster and easier to set up, a 2D–LC approach allows for a more accurate result with better selectivity and sensitivity. For both instrumental approaches, the signal ratio of the isotopologues was crucial and had a dominant effect on the results and the measurement uncertainty. For this reason, the different contributions to the measurement uncertainty were determined experimentally using Matched Standard Addition–IDMS and compared to the Exact Matching Double IDMS.

## 1. Introduction

Chemical contaminants in food and feed are of great public concern due to their potential health threats. The animal feed scandal in 2007 and the milk scandal in the following year were significantly severe food safety incidents. Especially in the latter scandal, the intentional adulteration with melamine (MEL) and cyanuric acid (CYA) did not only lead to a non–negligible number of deaths of animals and humans, but also to food recalls and widespread public outrage. As long–term consequence, intensified analytics regarding food contaminants in dairy products were intended. Before those scandals, only chemists dealing with plastics analysis knew about these industrial chemicals and their use, but today, all kinds of food scientists must deal with adulteration of sources of non–protein nitrogen. Both compounds are still in use in significant amounts as bulk chemicals, whereby MEL formaldehyde is used as raisin for the fabrication of laminates, plastics, glues, tableware, etc. [1]. CYA is a byproduct of the industrial use of melamine. It is used as bleaches, disinfectant, and is most known as a stabilizer for chlorine in outdoor pools [2]. Both compounds are chemicals with a high nitrogen content, which can induce misinterpretation of data from non–specific total protein measurement methods such as Kjeldahl analysis.

MEL and CYA are water–soluble compounds, whereby MEL shows a water solubility of 3.24 mg/mL H_2_O [3] and CYA shows a water solubility of 2.59 mg/mL H_2_O at 25 °C [4]. However, in concentrations exceeding 2 µg/mL, MEL and CYA crystallize in a one–to–one ratio to form melamine cyanurate, a very poorly water–soluble complex [5]. Several toxicology studies found evidence that the poorly soluble complex of MEL–CYA can cause kidney failure in humans and animals [6,7].

Due to the increasing intensity and severity of food fraud in the past years, diverse analytical methods have been developed and reported for MEL and its analogous CYA, whereas lots of them use sample clean-up preparation to remove matrix compounds, potentially disturbing analysis. Several sample preparation methods use and recommend SPE (solid phase extraction) [5,8,9,10]. Because of complex matrices and to protect the sensitive mass spectrometric (MS) systems, some working groups are using alternatively LC–MS/MS [8] and GC–MS [8,11], also immunoassays [12] or HPTLC methods [13]. 

Trace analysis in food matrices is very challenging. Here, multidimensional separation techniques are currently the state of the art. The 2D–LC method is claimed to be a more powerful tool regarding peak capacity and sample complexity, further enabling the mass spectrometer to analyze more compounds with high sensitivity [14]. Furthermore, a 2D–LC system can be used to overcome matrix effects that might interfere separation or the following MS ionization. 

To minimize matrix effects, the National Institute of Standards and Technology (NIST) uses techniques based on isotope dilution mass spectrometry (IDMS), which can offer further advantages to overcome matrix effects due to the similar behavior of the stable isotopes and analyte in sample preparation, extraction, chromatography, and MS ionization [15]. In general, IDMS is applied preferably when the accuracy of the results is of predominant analytical importance [16]. Especially the Exact Matching Double IDMS (EMD–IDMS) technique eliminates instrumental biases and allows for a precise measurement of the amount of substance in a sample against a similar reference material [17,18,19]. 

Despite instrumental effects on the measurement of the isotopologue amount, ratio differences in the sample matrix itself can also lead to biases. To reduce the matrix effect, MSA–IDMS can be introduced as a combination of standard addition and IDMS (Figure 1), which was first presented by Pagliano and Meija [20] and improved a couple of years later by Brauckmann and co–workers [21]. 

Although several MS methods are proposed for the analysis of MEL and CYA in different matrices, there is still a need for high accuracy methods especially for developing certified reference materials (CRM) [8,22]. The aim of the present study was to develop a fast–screening method for MEL and CYA. Beneath the one–point calibration as single IDMS, additional sophisticated methods are used, like the combination of standard addition and a double IDMS as described by Brauckmann and co–workers [21]. The adoption of MSA–IDMS has been attempted for the analysis of organic analytes. For comparison, results and uncertainty estimations were evaluated with the classic approach of EMD–IDMS as well as MSA–IDMS. All samples and calibration blends were prepared gravimetrically in a controlled environment with traceable temperature, relative air humidity, and air pressure [23]. Further, this method should be verified through participation in The Food Analysis Performance Assessment Scheme (*FAPAS*^®^) program of the UK, operating under the Food and Environment Research Agency (*FERA*) of the UK, to gain ISO/IEC 17025:2017 accreditation for both chemical compounds and especially infant formula as a matrix of interest. 

## 2. Materials and Methods

### 2.1. Reference Materials

The non–isotopically labeled CRM of melamine (2,4,6–triamino–1,3,5–triazine), MEL, was purchased from Merck KGaA (Buchs, Switzerland), while cyanuric acid (2,4,6–triol–1,3,5–triazine), CYA, was purchased from Dr. Ehrenstorfer™ (LGC Standards GmbH, Wesel, Germany). The isotopic labeled materials were both purchased from Merck KGaA (Buchs, Switzerland). The non–isotopically labeled homologous MEL has a chemical purity of 0.9995 g/g. The CYA CRM was certified with a content 0.983 g/g. The isotopically labeled homologous ^13^C_3_–MEL has a chemical purity of 0.988 g/g. The chemical purity of ^13^C_3_–CYA was stated with 0.996 g/g. Details about the distribution of the isotopologues were not given. 

For HPLC analysis, individual stock solutions of MEL, CYA, and their isotopically homologous ^13^C_3_–MEL and ^13^C_3_–CYA were prepared at 1 mg/mL in water by adding 6% (60 mL/L) formic acid. The individual working solution was obtained at 0.01 mg/mL by further dilution with acidified water. All solutions were stored in the dark at −20 °C. In contrary to MEL, CYA is less water–soluble; for this reason, CYA and MEL were dissolved by sonication for 30 min in the volumetric flask. HPLC–grade ammonium formate, acetonitrile, formic acid (0.99 g/g), and water were purchased from Merck KGaA.

### 2.2. Preparation of MEL and CYA in Infant Formula as an In–House Matrix Reference Material

CRM producers and laboratories need to participate in interlaboratory comparisons for their accreditation according to ISO/IEC 17025. In this context, proficiency testing (PT) vendors provide laboratories with specific samples contaminated with an appropriate amount of the requested chemical substances added to the matrix [19]. The item code for the PT sample in this study was 30,110 with a size of 50 g. In general, this PT round for 2021 was announced for food manufacturers and testing laboratories, whereby 22 participants were subscribed for CYA and 39 participants were subscribed for MEL.

In the present study, a pure solution containing a known amount of CYA, ^13^C_3_–CYA, and MEL, ^13^C_3_–MEL, respectively, was prepared as reference sample in the context of double IDMS. The preparation of a reference sample for double IDMS resulted in smaller uncertainties than single IDMS [24,25,26]. However, to better understand the influence of the matrix on the measured result, a self–made in–house matrix reference material with a known amount of CYA and MEL was prepared.

Hereby, the in–house matrix reference material was obtained by spiking the analytes at the beginning of the sample preparation to compare chromatographic peculiarities regarding matrix effect and peak shape with those of solvent–based curves. Therefore, 1 kg of infant formula was purchased from a local market and analyzed by an accredited LC–MS/MS method to confirm that it was free of MEL and CYA. An initial powder to be used as working standard with a concentration of 4.3 mg/kg for MEL, and 3.8 mg/kg for CYA, respectively, was gravimetrically prepared by dissolving an appropriate quantity of the neat material in water by 38 °C. An accurately weighed aliquot of standard solution was added into the infant formula by stirring for 30 min at 4 °C to ensure a homogenous distribution of MEL and CYA in the infant formula. The sample was lyophilized and ground to a fine powder with a blender. The mixed powder samples were dispensed into clean amber bottles and immediately capped with lids. The final product was stored at room temperature before analysis.

### 2.3. Sample Preparation for LC–MS Analysis

A homogenized test portion of 0.5 g infant formula was weighed into a 15 mL polypropylene centrifugal vessel. An extraction solvent (8 mL) of 50% (*V*/*V*) aqueous acetonitrile containing 6% (*V*/*V*) formic acid was added to extract the maximum amount of MEL and CYA. Protein precipitation was performed by adding acetonitrile as part of the extraction and precipitation solution. An aliquot of ca. 100 µL of each sample ^13^C_3_–MEL and ^13^C_3_–CYA with a concentration of 0.01 mg/g was added to the extraction solvent. Herein, the appropriate quantities of ^12^C–MEL and ^13^C_3_–MEL, as well as ^12^C–CYA and ^13^C_3_–CYA, respectively, should have a mass ratio as similar as possible. The vessel was placed in an ultrasonic bath for 30 min, mixed for another 30 min using an overhead shaker, and centrifuged at RCF = 12,000× *g* for 20 min (Labofuge™ 400, Fisher Scientific AG, Reinach, Switzerland). The supernatant formed after centrifugation was filtered through a 0.45 µm Filter (PP w/GMF, Whatman™, Huberlab AG, Industriestrasse 123, Aesch, Switzerland) and transferred to vials for LC–MS analysis.

### 2.4. 2D–HILIC–ESI–MS/MS Analysis

In the 2D chromatography, only a part of the peak in 1D is cut out and running over a second column, whereby less matrix contaminants enter the MS system. Furthermore, just minor sample pretreatment can be applied as well, which is crucial for a precise determination of the signal ratios with a large set of samples and multiple injections. Crude infant formula solutions were analyzed directly by 2D–HILIC–ESI–MS/MS. Calibration curves were obtained by injecting different concentrations of the standards, and the areas obtained were used to plot a linear curve. For quantitative analysis, linear regression was used (*r*^2^ > 0.999 and *N* = 4). 

Chromatographic separation was achieved using an Agilent 1260 Series LC system (Agilent Technologies Inc., Waldbronn, Germany) coupled with ESI–MS/MS. Due to their hydrophilic character, a hydrophilic interaction liquid chromatography (HILIC) column was used. For 1D, the analysis was performed on a TSKgel^®^ amide 80 (150 mm, 3 mm ID, particle size 3 µm, Tosoh Bioscience GmbH, Griesheim, Germany) and for 2D on a TSKgel^®^ amide 80 (20 mm, 2 mm ID, particle size 3 µm, Tosoh Bioscience GmbH, Griesheim, Germany), whereby both were maintained at 40 °C. 

The 1D separation was achieved with a mobile phase consisting of (A) 5 mmol/L ammonium formate in water (pH 3.0) and (B) acetonitrile and at a flow rate of 0.4 mL/min with a gradient elution employed with the ratio of (A) and (B) varied as follows: 25% A (0–4 min), 20–95% A (4–4.2 min), 95% A (4.2–7 min), 95–25% A (7–7.1 min), and a re-equilibration with 25% A (7.1.1–11.5 min) (≈ 4 column void volumes). Sample injection volume was 1 µL. CYA was eluted under these conditions at 2.47 min and MEL at 4.14 min. Both compounds were cut out for the 2D separation with a sampling time of 0.1 min and a loop filling of 40 µL.

In the present study, the same mobile phase was used for 1D as well as for 2D, but with the following gradient for the solvents (A) and (B): 40% A (0–0.4 min), 40–90% A (0.4–0.6 min), 90% A (0.6–1.4 min), and 40% A (1.4–2.8 min). 

The MS detection was performed in multiple reaction monitoring (MRM) mode. MRM transitions, optimized collision energy (CE), and fragmentor (V) parameters are listed in Table 1. Shortly, mass detection was set at *m/z* 127 → 85 for ^12^C–MEL and *m/z* 130.1 → 86.9 for ^13^C_3_–MEL with positive electrospray ionization mode, respectively, at *m/z* 128 → 42 for ^12^C–CYA and *m/z* 131 → 43.1 for ^13^C_3_–CYA with negative electrospray ionization mode. Nitrogen was used as curtain, nebulizer, and collision gas. The source parameters including gas temperature, gas flow, nebulizer, sheath gas temperature, sheath gas flow, and nozzle voltage were set to 220 °C, 6 L/min, 40 psi, 330 °C, 10 L/min, and 1800 V, respectively. MassHunter Workstation (Quantitative Analysis for QQQ 10.1, Agilent Technologies Inc., Waldbronn, Germany) software was used for system control, data collection, and processing.

In general, the chromatographic sequence was set up as followed including the sample with spiked IS, the reference sample with labeled and unlabeled MEL and CYA sample including the IS spike and procedural blanks. For every 10 samples, HPLC grade water was injected into LC–MS/MS to check for carry–over of target chemicals between samples.

### 2.5. Method Validation 

The Eurachem Method Validation Guideline (Eurachem/CITAC) by Ellison, Roesslein, and Williams [27] is the basis for the evaluation process of the LC–MS/MS methods with respect to selectivity, repeatability, linearity, sensitivity, and robustness. The concept behind the method validation is based on a single stock solution of the analyte for both parts, sample and reference. The spike solution for all blends is also from one stock solution. As a result, the content or concentration is kept constant and all biases and contributions to the measurement uncertainty can be detected and measured.

Concerning the selectivity, both isotopologues showed no overlap in the monoisotopic signals of the analytes, which is also crucial for the derivation of the IDMS equation. The corresponding equations are described in the following sections. 

This concept is applied for EMD–IDMS and MSA–IDMS, allowing for the determination of their performance characteristics including the trueness and precision of the result including the sample preparation procedure and weighing, the precision of the signal ratio, and intermediate precision. The concept also allows for the differentiation between single contributions to the measurement uncertainty, e.g., instrumental bias, repeatability, sample preparation, as well as the influence of matrix. All binary and ternary blends were prepared from the same CRM and spike stock solutions, so that the content in sample and reference is the same (*w*_X_ = *w*_Z_). In that case, the target value of Equations (1) and (8) becomes dimensionless. Deviations from the average value of a measurement to this target value can be referred to as biases. The standard deviation of the repeated measurement of one blend is used to determine the uncertainty contribution of the instrument *u*(inst), and the repeated measurement of the whole set of blends yields the uncertainty contribution of instrument and preparation *u*(inst) + *u*(prep), which also covers biases, e.g., undetected sample loss or insufficient removal of static charges. The validation of the robustness focuses on a possible bias caused by non–matched samples. Thus, two repetitions of non–matched standard addition experiments were conducted with the PT 30110 sample (*FERA/FAPAS*^®^). The same sample was analyzed also with the EMD–IDMS method as part of the corresponding PT by *FAPAS*^®^, allowing for the cross–validation and comparison of the two different and partly new methods.

## 3. Results

### 3.1. Exact Matching Double IDMS (EMD–IDMS)

The aim of the present study was to develop a high–precision method for the determination of MEL and CYA in matrix samples. This method should be verified through participation in the UK–based *FAPAS*^®^ program, operating under the *FERA* of the UK, to gain accreditation according to ISO/IEC 17025 for both chemical compounds and especially infant formula as a matrix of interest. The results and uncertainty measurement were evaluated with the classic approach of an EMD–IDMS as well as the MSA–IDMS.

IDMS is considered as a metrologically traceable method concerning the International Systems of Units (SI) [28]. The EMD–IDMS quantification in the present study bases on the equation derived by Henrion [29] and by Sargent, Harrington, and Harte [18]. Therefore, to achieve an accurate and precise measurement, all preparations were conducted gravimetrically according to metrological weighing procedures [30]. 

However, one should bear in mind the differing definitions of the isotope amount ratio, as they define the calculation of the signal ratios of analyte with natural isotopic composition (*A*_1_) and its labeled isotopologue (*A*_2_).
(1)$$w_{X}=w_{Z}\cdot \frac{m_{Y}}{m_{X}}\cdot \frac{m_{Zc}}{m_{Yc}}\cdot \frac{R_{B}}{R_{Bc}}$$

*w*_X_ = mass fraction of analyte in sample X (µg/g);

*w*_Z_ = mass fraction of analyte in spike Y (mol L^−1^);

*m*_Y_ = mass of spike Y added to the sample X to prepare the blend B (mg);

*m*_X_ = mass of sample X added to the spike Y to prepare the blend B (mg); 

*m*_Zc_ = mass of primary standard solution Z added to the spike Y to make calibration blend B_C_ (mg);

*m*_Yc_ = mass of spike Y added to the primary standard solution Z to make calibration blend B_C_ (mg); 

*R*_B_ = isotope amount ratio of sample blend B;

*R*_Bc_ = isotope amount ratio of calibrant blend B_C_.

As described by Sargent, Harrington, and Harte [18], the isotope amount ratio is defined by Equation (2):(2)RB=nXE1+nYE1nXE2+nYE2~A1A2

*n*_X,Y_ = amount of analyte ^1^E (MEL, CYA) and isotopologue ^2^E (^13^C_3_–MEL, ^13^C_3_–CYA) in sample X and spike Y;

*A*_1_, *A*_2_ = signal areas of analyte or reference (*A*_1_) and the isotopologue or monitor (*A*_2_).

As discussed in other publications, the contributions to the measurement uncertainty can be referred to the weighing of the sample, contributions of reference and spike solutions, and the measurement of the signal ratios of sample–spike and reference–spike mixtures [29,31]. Usually, the major contribution arises from the signal ratios, which is influenced by the ionization process, chromatographic system, and effects by the sample matrix itself. The analysis of the measurement uncertainty for the EMD–IDMS measurements according to the Eurachem/CITAC guide and the NIST calculation tool according to Ellison, Roesslein, and Williams [27] is presented in Table 2. As the two mixtures are measured in a bracketed sequence, the standard deviation of the signal ratio and the ratio of the signal ratios *R*_B_/*R*_Bc_ are calculated from each subsequent pair. Each pair is injected repeatedly. 

IDMS requires a linear response of analyte and isotopologue. The signal area ratios *A*(^13^C_3_–MEL)/*A*(MEL) or *A**/*A* versus the corresponding mass fractions *m*(^13^C_3_–MEL)/*m*(MEL) or *m**/*m* results in a straight line through the origin with a defined slope, indicating no interferences or instrumental biases between labeled and unlabeled compounds [32]. The evaluation of the signal areas as a function of weighing ratios shows a straight line with *r*^2^ > 0.999 (Figure 2). The area ratio of CYA/^13^C_3_–CYA and MEL/^13^C_3_–MEL in the calibration blend *R*_BC_ and the sample blend *R*_B_ should be within 0.95–1.15 and their corresponding ratio *R*_B_/*R*_Bc_ should be close to unity for minimum measurement uncertainty and instrumental bias. 

The standard deviations of the signal ratios *s*(*R*_B_/*R*_Bc_) and of Equation (1) *s*(*w*_X_/*w*_Z_) with *w*_X_ = *w*_Z_ are used to determine the repeatability and robustness (intralab, interday). Biases can be calculated from the difference between the mean value and the target value of *w*_X_/*w*_Z_ = 1. As the signals of the analytes MEL and CYA were separated by chromatography and the measurement of the pure solutions did not indicate any interferences, additional biases must stem from the matrix itself or fractionation caused by overload of the HPLC columns.

In the present study, CYA/^13^C_3_–CYA and MEL/^13^C_3_–MEL were used. However, with regard to matrix removal and matrix effect, a heart–cut 2D–LC method was developed and compared to the 1D–LC. For the 2D–LC, the retention time (RT) of the MS signals must not vary more than 0.05 min. Otherwise, the system must be equilibrated longer. Furthermore, the signal area of CYA and MEL in 2D should be constant. Figure 3 shows the chromatogram(s) of the PT 30110 sample (*FERA/FAPAS*^®^) 30110 extract spiked with ^13^C_3_–CYA and ^13^C_3_–MEL with an RT of 2.8 min for CYA and RT 4.4 min for MEL in 1D, and an RT of 4.0 min for CYA and RT 5.7 min for MEL in 2D. The analysis time nearly doubled for the LC × LC. However, a negligible slope deviation was observed with low impact on the resulting signal ratios. 

As shown with the boxplot analysis for the isotope ratio (Figure 4), separations in 1D and 2D for both compounds are not in accordance with each other, whereby a lower scattering was observed for the 2D. This can be explained by the increasing resolution for 2D as a result of the online sample clean-up in 1D. This result is in accordance with the study described by Breidbach and Ulberth [31], who observed identical medians for their 1D and 2D approaches, but with a smaller dispersion for the LC × LC. According to them [31], the signal of the 2D was much higher and therefore had less scattering. However, although the 2D displays a certain asymmetry especially for the MEL peak, it is very stable throughout the sequence.

The distribution of all single results of the present study are shown as Monte Carlo Simulation (MCS) in Figure 5. EMD–IDMS yielded the following content for infant formula with 12.6 mg kg^−1^ ± 1.4 mg kg^−1^ for CYA and 12.2 mg kg^−1^ ± 1.0 mg kg^−1^ for MEL (*k* = 2). The certified value of the PT 30110 sample (*FERA/FAPAS*^®^) was set to 11.8 mg kg^−1^ for CYA and 12.4 mg kg^−1^ for MEL, respectively. 

### 3.2. Matched Standard Addition–IDMS (MSA–IDMS)

Based on the standard addition–IDMS technique [21], the present study also used a procedure for the analysis of organic molecules, where the corresponding isotopologue (“spike”) has no detectable overlap with the signals of the analyte molecule. This is usually the case for mass differences of 3 Da or more and a high content of the “spike” isotopologue (>97% (g/g)). If an overlap occurs, it is necessary to characterize the “spike”, “sample”, and “reference” solutions individually and use Equation (2) with the individual contributions to the isotope/isotopologue amount ratios for blend *R*_b*,i*_, sample *R*_x_, and spike *R*_y_ as described in Brauckmann [21]:(3)$$\underset{=y_{i}}{\underbrace{\frac{m_{y,i}}{m_{x,i}}\cdot \frac{R_{y}-R_{b,i}}{R_{b,i}-R_{x}}}}=\underset{=a_{1}}{\underbrace{\frac{1}{w_{Y}}\cdot \frac{M_{y}}{M_{x}}\cdot \frac{\sum R_{y}}{\sum R_{x}}\cdot w_{Z}}}\cdot \underset{=x_{i}}{\underbrace{\frac{m_{z,i}}{m_{x,i}}}}+\underset{=a_{0}}{\underbrace{\frac{1}{w_{Y}}\cdot \frac{M_{y}}{M_{x}}\cdot \frac{\sum R_{y}}{\sum R_{x}}\cdot w_{X}}}$$

*m*_y*,i*_ = mass of “spike” (y) in blend *i* (g);

*m*_x*,i*_ = mass of sample (x) in blend *i* (g);

*m*_z*,i*_ = mass of “reference” (z) in blend *i* (g);

*R*_y_, *R*_b*,i*_, *R*_x_ = isotope/isotopologue amount ratio in “spike” (y), blend *i* (b,*i*), sample (x) (mol/mol);

*w*_Y_, *w*_Y_, *w*_Z_ = mass fraction of the analyte in “spike” (y), sample (x), “reference” (z) used to prepare the blends b,*i* (g/g).

Since there are no overlapping contributions of the “spike” to the amount ratios of “sample” or “reference”, the “eq. A12” in the manuscript by Brauckmann et al. [21], Equation (3), simplifies to Equation (4) under the constraints that the amount fraction of the target analyte (denoted as 1) in the “spike” is *x*_y1_ = 0 and the amount fraction of the “spike” (denoted as 2) in the sample is also *x*_x2_ = 0. Equation (5) represents the linearized form with the dimensionless *y*-axis values of the ternary mixtures *i*, the corresponding *x*-axis values, and the definition of slope *a*_1_ and intercept *a*_0_.
(4)$$\frac{m_{y,i}}{m_{x,i}}\cdot \frac{x_{y2}-R_{b,i}\cdot x_{y1}}{R_{b,i}\cdot x_{x1}-x_{x2}}=\frac{1}{w_{Y}}\cdot \frac{M_{y}}{M_{x}}\cdot w_{Z}\cdot \frac{m_{z,i}}{m_{x,i}}+\frac{1}{w_{Y}}\cdot \frac{M_{y}}{M_{x}}\cdot w_{X}$$
(5)$$\Leftrightarrow \frac{m_{y,i}}{m_{x,i}}\cdot \frac{x_{y2}}{R_{b,i}\cdot x_{x1}}=\frac{1}{w_{Y}}\cdot \frac{M_{y}}{M_{x}}\cdot w_{Z}\cdot \frac{m_{z,i}}{m_{x,i}}+\frac{1}{w_{Y}}\cdot \frac{M_{y}}{M_{x}}\cdot w_{X}$$
(6)$$\Leftrightarrow \underset{=y_{i}}{\underbrace{\frac{m_{y,i}}{m_{x,i}}\cdot \frac{1}{R_{b,i}}}}=\underset{=a_{1}}{\underbrace{\frac{1}{w_{Y}}\cdot \frac{M_{y}\cdot x_{x1}}{M_{x}\cdot x_{y2}}\cdot w_{Z}}}\cdot \underset{=x_{i}}{\underbrace{\frac{m_{z,i}}{m_{x,i}}}}+\underset{=a_{0}}{\underbrace{\frac{1}{w_{Y}}\cdot \frac{M_{y}\cdot x_{x1}}{M_{x}\cdot x_{y2}}\cdot w_{X}}}$$

One should keep in mind the difference in the definition of the blend’s isotopologue amount ratios *R*_b*,i*_ (see “eq. A1” written in the manuscript by Brauckmann et al. [21] et al. to the EMD–IDMS equation above, which is still the sum of the amounts of monitor (spike) denoted as 2 and the reference (target analyte) denoted as 1, but is now the inverted signal ratio *R*_b*,i*_ = *f*_inst_·*A***_2_**/*A***_1_**.
(7)$$R_{b,i}=\frac{n_{x2,i}+n_{z2,i}+n_{y2,i}}{n_{x1,i}+n_{z1,i}+n_{y1,i}}=f_{inst}\cdot \frac{A_{2}}{A_{1}}$$

As the instrumental biases *f*_inst_ are part of slope and intercept, they are eliminated when Equation (7) is derived from Equation (5), which is used to calculate the content of the analyte in the sample *w*_X_. Additive biases from the instrument were excluded during the validation of the method. They should be detectable by repeated injection of the reference–spike mixture.
(8)$$w_{X}=\frac{a_{0}}{a_{1}}\cdot w_{Z}$$

For the standard addition experiment, one binary mixture of sample and spike and three–to–four ternary mixtures of sample, spike solution, and reference solutions were prepared. The first mixture of sample and spike solution represents the intercept with the *y*-axis. When possible, this mixture should be matched in the signal ratio close to 1 for minimizing measurement uncertainty. For the ternary mixtures, the spike and reference solution were added to a constant amount of sample with the same ratio to keep the signal ratios constant, which is crucial for minimum instrumental biases and the validity of the calculations. 

For the method validation of the MSA–IDMS technique, four blends were prepared gravimetrically using dispenser pipettes to guarantee a matching of all signal ratios. Again, one stock solution was prepared of the analytes MEL and CYA once and represents both the sample and reference solution to eliminate the content of sample and reference in Equation (8). In that case, slope *a*_0_ and intercept *a*_1_ should have the same values to yield the target value of *w*_X_/*w*_Z_ = 1. Deviations from this target value indicate a systematic bias of instrument or sample preparation and can be used as contribution to the measurement uncertainty. Repeatability and biases can be determined from the standard deviation of the signal ratios (*N* = 3) of repeated injections and the deviation from the target value from repeated experiments. 

All blends need to be measured in sequence over a much longer period compared to the bracketing technique used for the EMD–IDMS measurement, which makes all standard addition approaches more prone to instrumental drift and higher influences of outliers on the resulting content of the sample and less robust compared to EMD–IMDS. Both effects, instrumental drift and outlier, have a significant influence on the slope and intercept of the regression analysis if the number of mixtures (<4) and measurements is too low (<3). Calculating the measurement uncertainty from the regression may underestimate the contributions from these effects. Outliers should be tested by applying the whole calculation and MCS results in higher uncertainties and broader coverage intervals, as shown in Figure 6.

The results of the MSA–IDMS experiment for MEL and CYA of the CRM solutions without matrix are shown in Table 3. The mean value of <*R*_b,*i*_> for all four blends has a small standard deviation for MEL and CYA from 0.6 to 0.8%, indicating a good matching for all blend preparations. The experimental relative standard uncertainty of the average signal ratios *R*_b*,i*_ is 3.7% (MEL) and 1.7% (CYA) and agrees with the standard deviation from the MCS for *a*_0_/*a*_1_. This also fits to the usual repeatability of the signal ratio measurement of these 2 analytes ranging from 2.0 to 3.0% (RSD). 

Finally, additional contributions of the matrix to the measurement uncertainty or to a possible bias were evaluated by individual MSA–IDMS measurements of the PT 30110 sample (*FERA/FAPAS*^®^) with unmatched blends, which should simulate deviations from the above assumption in the equations, as it was also tested for EMD–IDMS [33]. Due to low signal intensities for CYA, reasonable results could only be achieved for MEL. The mean value of the two measurements for *w*_X_(MEL) = 12.5 mg/kg with a standard deviation *s*(*w*_X_(MEL)) = 0.4 mg/kg. This is in good agreement with the assigned value for MEL in this PT sample of 12.4 mg/kg (see PT 30110 report (*FERA/FAPAS*^®^)) and the value from EMD–IDMS measurements of the same material of 12.2 mg/kg (*U*(MEL) = 1.0 mg/kg, *k* = 2).

### 3.3. Uncertainty Estimation of EMD–IDMS and MSA–IDMS 

The combined measurement uncertainty of the EMD–IDMS and MSA–IDMS experiments was estimated according to the GUM [34] and the Eurachem/CITAC guide [27]. The uncertainties of both techniques were determined by repeated measurements of CRM solutions, in–house produced reference material, and an external PT material to compare both techniques under the influence of a sample matrix and the precision of the measurement. 

MCS is conducted using Microsoft Excel to calculate the mean and median values, standard deviations, and coverage intervals (95% confidence) for the content *w*_X_ with respect to Equations (1), (6) and (8). Equation (1) serves as model for EMD–IDMS experiment and Equations (6) and (8) for the MSA–IDMS experiment performed by Ellison, Roesslein, and Williams [27]. The online tool “Uncertainty Machine” by NIST can also run MCS and deliver more precise results with larger numbers of data points compared to Excel but only for the EMD–IDMS experiments. It is based on R core [35] but allows only a single model equation and one output quantity. Finally, the software GUM Workbench Pro (version 2.4.1.406, Metrodata GmbH, Germany) is used to estimate the combined measurement uncertainty of Equations (6) and (8) including the nested linear regression analysis for slope *a*_1_ and intercept *a*_0_. 

### 3.4. Uncertainty of EMD–IDMS of MEL/^13^C_3_–MEL of the PT Sample 

The combined measurement uncertainty is calculated for a typical measurement of MEL in the PT 30110 sample (*FERA/FAPAS*^®^). The results of the PT will be compared to the MSA–IDMS results in a matrix to validate the methods performance. MCS and linear GUM approach are calculated within the online NIST tool “Uncertainty Machine” (Table 3).

### 3.5. Uncertainty Determination of MSA–IDMS of Neat MEL/^13^C_3_–MEL Solution in Reference Matrix

MCS was applied for the analysis of a single sample set of pure aqueous MEL solutions to compare both approaches. Table 4 shows the definition of all single measurands for the four blends including their uncertainties *u*, sensitivity coefficients *c*, and indices of MEL. 

Like in the EMD–IDMS experiment, the standard deviation of the signal ratio 1/*R*_b*,i*_ dominates contributions to the uncertainty budget. The simulation is calculated with 10,000 data points due to the slow calculation of the linear regression parameters within Excel. Figure 7 shows the resulting distribution of the results for *w*_X_/*w*_Z_ for CYA and MEL.

Table 5 shows the definition of all single measurands for the four blends including their uncertainties *u*, sensitivity coefficients *c* and indices of CYA.

After the measurement of the CRM solutions without matrix, the standard addition technique was applied also to the PT 30110 sample (*FERA/FAPAS*^®^). The number of repeats is smaller for this experiment. Thus, a relative standard uncertainty *u*_rel_(*R*_b_) = 2.0% is assumed (Table 6) to allow for a comparison to the results in Table 2 and the results of the GUM Workbench program in the Table 7. 

Contrary to the previous MCS results of the pure solutions, an asymmetric distribution is observed, which also leads to differences between the mean value of the simulated results, calculated result of the measurement, and the percentiles of the MCS distribution (Figure 8). 

## 4. Conclusions

A new standard addition method for IDMS for the analysis of MEL and CYA in infant formula based on the previously published method by Brauckmann et al. (2021) was presented [21]. Additionally, a validation scheme with stock solutions of pure neat and matrix CRM allows for the detailed evaluation of single contributions to the combined measurement uncertainty according to the Eurachem/CITAC guidelines (Ellison, Roesslein, and Williams, 2012, [27]). 

As previously mentioned, the melamine–cyanurate complex is somewhat more toxic than each chemical alone, whereby the limits for melamine in powdered infant formula is defined by 1 mg/kg (WHO, 2009) [36]. By using the 2D–LC method, a simple and fast sample preparation can be considered, ensuring a smaller error bar than the more commonly used 1D–LC method.

The present study showed that EMD–IDMS and MSA–IDMS were both effective quantitative methods with similar quantitative results indicating that the quantification of MEL and CYA in infant formula is barely affected by matrix. Both methods could be used to assign values for mass fractions of these analytes in matrix–based CRM or test material for inter–laboratory PT schemes. A comprehensive approach in the evaluation of the measurement uncertainty including Monte Carlo simulations was applied to allow a detailed picture about the main contributions to uncertainty and sources of experimental errors, e.g., precision and robustness of the signal ratios, instrumental drift during MSA–IDMS measurements, and sufficient number of mixtures depending on the desired precision of the sample mass fraction *w*_X_. 

Moreover, in these days, the addition of chemical by–products like MEL or CYA can be considered as a real threat not only in infant formula but also in alternative proteins, for example, in the future–oriented meat sector. In summary, an LC–IDMS method for the quantification of MEL and CYA in infant formula was developed that shows to be precise and exact, which is not affected by analyte loss or sample carry–over effect, and can be used as a high throughput method. The sample preparation is simple, so that in general this method went from a reference method to a routine method and can be used as a routine laboratory method. The method meets all the requirements of European regulations regarding the analysis of MEL and CYA, and the quality of the results obtained is supported by accreditation according to ISO/IEC 17025.

## Figures and Tables

**Figure 1 foods-13-02377-f001:**
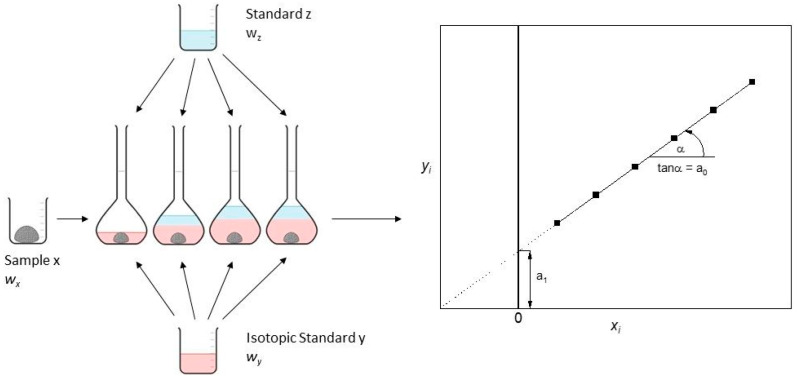
The principle of MSA–IDMS measurement is shown.

**Figure 2 foods-13-02377-f002:**
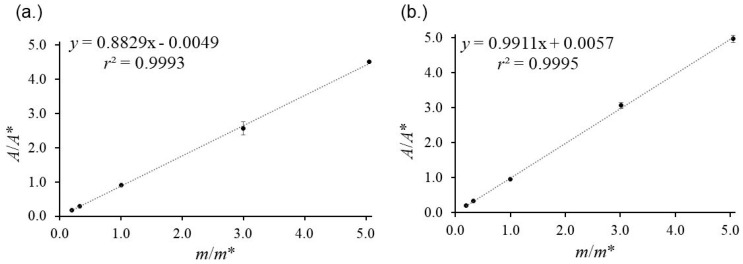
The linear calibration curve of analyte and isotopologue. Calibration curves are calculated with individual (**a**) ^12^C_3_–CYA and ^13^C_3_–CYA mixtures; (**b**) ^12^C_3_–MEL and ^13^C_3_–MEL with five *m**/*m* ratios. MEL and CYA are in a high linear working range.

**Figure 3 foods-13-02377-f003:**
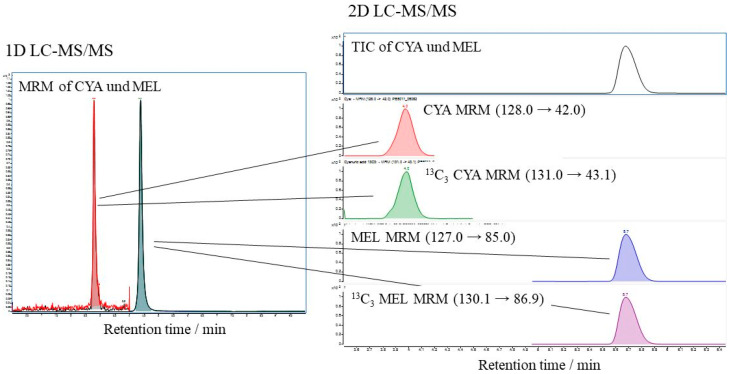
Visual comparison of 1D–LC and 2D–LC implementation, whereby heart–cut LC × LC was used where one single fraction of the 1D column is injected onto the 2D column.

**Figure 4 foods-13-02377-f004:**
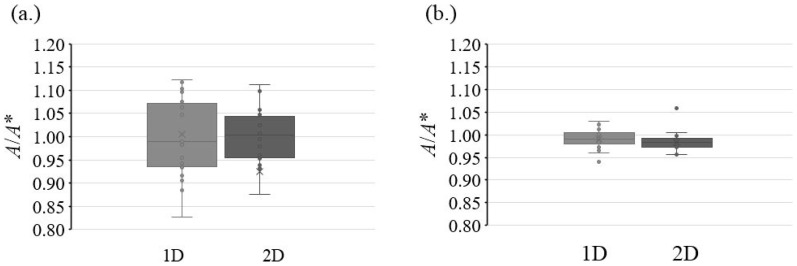
The repeatability was analyzed by repeated injection of the same sample (*N* = 12). A boxplot representation for CYA (**a**) using 1D–LC and MEL (**b**) using 2D–LC shows the difference between the two methods. The black line in the middle of the boxplot depicts the mean of the two ion ratios. Since *A*/*A** is proportional to *R_B_* and *R_Bc_*, the repeatability is a crucial factor in the combined measurement uncertainty.

**Figure 5 foods-13-02377-f005:**
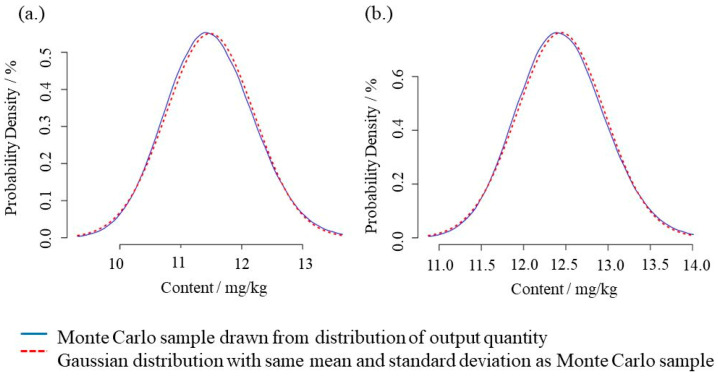
Distribution of output quantity calculated from the MCS results of the NIST Uncertainty Machine for CYA (**a**) and MEL (**b**) of the EMD–IDMS measurements of the PT 30110 sample (*FERA/FAPAS*^®^, 2021) using Equation (1) and the parameter set in Table 2 for the NIST Uncertainty Machine. Due to the symmetric Gaussian distribution of the simulated results, the mean values and standard deviations do not differ significantly from median and coverage interval with 95% probability. The other numerical results are presented in Table 3.

**Figure 6 foods-13-02377-f006:**
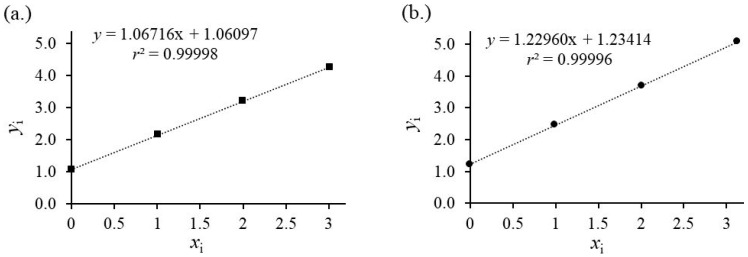
(**a**) Standard addition results of the binary and ternary mixtures of one CYA and ^13^C_3_–CYA stock solution using Equation (6) for the calculation of *y_i_* and *x_i_*. Linear regression shows a very good fit. Slope and intercept result in a good agreement with the target value *w*_X_/*w*_Z_ = 0.994 with *u*(*w*_X_/*w*_Z_) = 0.009. Including the outliers leads to the result *w*_X_/*w*_Z_ = 0.979 with *u*(*w*_x_/*w*_Z_) = 0.021. A *t*-test for the comparison of the expected value 1 to the measured value results *t*(*n* = 8, α = 0.975) = 1.89 < 2.36. (**b**) Standard addition results of the binary and ternary mixtures of one MEL and ^13^C_3_–MEL stock solution using Equation (6) for the calculation of *y_i_* and *x_i_*. Linear regression shows a very good fit. Slope and intercept result in a good agreement with the target value *w*_X_/*w*_Z_ = 1.004 with *u*(*w*_X_/*w*_Z_) = 0.012. A *t*-test for the comparison of the expected value 1 to the measured value results *t*(*n* = 8, α = 0.975) = 0.94 < 2.36.

**Figure 7 foods-13-02377-f007:**
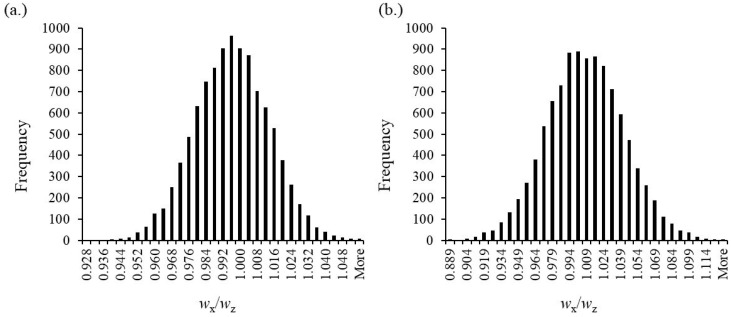
(**a**) Distribution of the simulated results for the case of a single CYA CRM stock solution representing sample x and reference z. The target value is 1. The coverage interval ranges from 0.961 to 1.030 with 95% confidence (*N* = 10,000). (**b**) Distribution of the simulated results for the case of a single MEL CRM stock solution representing sample x and reference z. The target value is 1. The average value is 1.005 with a coverage interval of 0.937–1.071 (95% confidence). A reasonable distribution with *N* = 10,000 results.

**Figure 8 foods-13-02377-f008:**
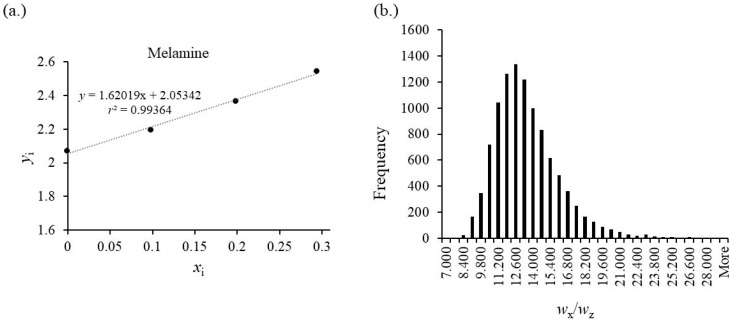
(**a**) Standard addition results of MEL in infant formula (PT sample (*FERA/FAPAS*^®^) 30110, 2021) using a CRM solution with *w*_Z_ = 10.08 mg/kg for the ternary mixtures. A lower precision caused by the effect of the matrix on the measurement of signal ratio and the smaller number of repeats (*N* = 2, total number *n* = 8) leads to a lower *r*^2^ value compared to the solutions. Slope and intercept result in a content of *w*_X_ = 12.8 mg/kg with an uncertainty *U*(*w*_X_) = 0.85 mg/kg according to Brauckmann et al. [21]. The assigned value of the CRM for MEL is *w*(MEL)*_CRM_* = 12.4 mg/kg +/–0.2 mg/kg (*k* = 2). Using a *t*-test (two-way, one sample), the results for the comparison with the MSA–IDMS measurement is a value of *t*(*n* = 8, α = 0.975) = 0.92 < 2.36. (**b**) Distribution of the mass fraction of MEL in the PT sample resulting from MCS with the parameters set in Table 2. The mean value of *w*_X_(MCS) = 13.2 mg/kg has a standard deviation of *s*(*w*_X_) = 2.6 mg/kg. The deviation from the experimental value *w*_X_ = 12.8 mg/kg originates from the asymmetric distribution with a tailing towards higher values. The calculation of the median and coverage interval with a probability of 95% results the following percentiles at 9.4 mg/kg (2.5 %), 12.8 mg/kg (50 %, median), and 19.4 mg/kg (97.5 %).

**Table 1 foods-13-02377-t001:** The optimized fragmentor and collision energy for each transition of the 2D–LC–MS.

Name	Precursor Ion [*m/z*]	Product Ion [*m/z*]	RT min	Ion Polarity	Collision Energy V	Fragmentor V
CYA	128.0	42.0	2.7	negative	12	50
^13^C_3_–CYA	131.0	43.1	2.7	negative	12	50
MEL	127.0	85.0	6.7	positive	16	113
^13^C_3_–MEL	130.1	86.9	6.7	positive	16	113

**Table 2 foods-13-02377-t002:** Calculation parameters for the MCS (NIST Uncertainty Machine) of a typical EMD–IDMS measurement of MEL in the PT 30110 sample (*FERA/FAPAS*^®^). Mean value and standard deviation of the gaussian distribution and their results are *w*_X_(MCS) = 12.44 mg/kg (*u*(*w*_X_(MCS)) = 0.52 mg/kg) and *w*_X_(GUM) = 12.43 mg/kg (*u*(*w*_X_(GUM)) = 0.52 mg/kg).

Measurand	Mean	*u*	Monte Carlo Simulation ANOVA (Relative Contributions in %)	GUM c*_i_*	GUM Index (*u_r_*^2^ in %)
*m*_X_/mg	500	0.144	0.01	−0.025	0.0047
*m*_Y_/mg	594	0.229	0.01	0.021	0.0085
*m*_Zc_/mg	595	0.172	0.01	0.021	0.0048
*m*_Yc_/mg	598	0.173	0.01	−0.021	0.0048
*R* _B_	1.02	0.025	34.31	12.000	34.0000
*R* _Bc_	0.97	0.030	54.37	−13.000	55.0000
*w*_Zc_/mg/kg	10.0	0.14	11.13	1.2	11.0000
Residual	–	–	0.16	–	–

**Table 3 foods-13-02377-t003:** Results of the MSA–IDMS experiment for MEL/CYA in solution including the standard deviations of the experimental data and the Monte Carlo simulation (MCS). They are similar to the results of the EMD–IDMS experiments with two blends from the same CRM and spike stock solutions. Target value for *a*_0_/*a*_1_ = 1 with *w*_Z_ = *w*_X_.

	MEL	CYA
<*R*_b,*i*_>	1.221 ± 0.008 (0.6%)	1.056 ± 0.008 (0.8%)
*u*_rel_ (*R*_b,*i*_)	3.7%	1.7%
*a*_0_/*a*_1_	1.003 ± 0.034 (MCS)	0.994 ± 0.017

**Table 4 foods-13-02377-t004:** Definition of the MCS for the MSA–IDMS with 4 blends including the binary blend 1. Masses of the MEL stock solution *m*_x_, *m*_z_ (*N* = 1) and the ^13^C_3_–MEL spike solution *m*_y_ (*N* = 1) are simulated for each standard addition blend 1–4. Sensitivity coefficients *c_i_* and the indices are calculated for experimental data 1/*R*_b*,i*_ (*N* = 3) according to (NIST/SEMATECH, 2012).

Measurand	Mean	*u*	*c_i_*	*c_i_*^2^·*u*_r_^2^/10^−5^	Index
*m*_x,1_/mg	48.510	0.049	1.00	0.10	0.2%
*m*_y,1_/mg	49.002	0.049	1.00	0.10	0.2%
*m*_z,1_/mg	0.000	–	–	–	–
*R* _b,1_	1.217	0.013	0.58	3.90	8.4%
*m*_x,2_/mg	49.523	0.050	1.00	0.10	0.2%
*m*_y,2_/mg	99.193	0.099	1.00	0.10	0.2%
*m*_z,2_/mg	49.053	0.049	1.00	0.10	0.2%
*R* _b,2_	1.231	0.038	0.58	32.20	69.4%
*m*_x,3_/mg	48.851	0.049	1.00	0.10	0.2%
*m*_y,3_/mg	147.450	0.147	1.00	0.10	0.2%
*m*_z,3_/mg	98.266	0.098	1.00	0.10	0.2%
*R* _b,3_	1.224	0.020	0.58	8.50	18.3%
*m*_x,4_/mg	47.388	0.047	1.00	0.10	0.2%
*m*_y,4_/mg	198.577	0.199	1.00	0.10	0.2%
*m*_z,4_/mg	148.252	0.148	1.00	0.10	0.2%
*R* _b,4_	1.213	0.005	0.58	0.67	1.5%

**Table 5 foods-13-02377-t005:** Definition of the MCS for the MSA–IDMS with 4 blends including the binary blend 1. Masses of the CYA stock solution *m*_x_, *m*_z_ (*N* = 1) and the ^13^C_3_–CYA spike solution *m*_z_ (*N* = 1) are simulated for each standard addition blend 1–4. Sensitivity coefficients *c_i_* and the indices are calculated for experimental data 1/*R*_b*,i*_ (*N* = 3) according to NIST/SEMATECH (2012).

Measurand	Mean	*u*	*c_i_*	*c_i_*^2^·*u*_r_^2^/10^−5^	Index
*m*_x,1_/mg	48.695	0.049	1.00	0.10	1.0%
*m*_y,1_/mg	48.943	0.049	1.00	0.10	1.0%
*m*_z,1_/mg	0.000	–	–	–	–
*R* _b,1_	1.051	0.007	0.58	1.28	12.3%
*m*_x,2_/mg	48.218	0.048	1.00	0.10	1.0%
*m*_y,2_/mg	98.725	0.099	1.00	0.10	1.0%
*m*_z,2_/mg	48.900	0.049	1.00	0.10	1.0%
*R* _b,2_	1.048	0.003	0.58	0.34	3.2%
*m*_x,3_/mg	49.055	0.049	1.00	0.10	1.0%
*m*_y,3_/mg	147.418	0.147	1.00	0.10	1.0%
*m*_z,3_/mg	98.247	0.098	1.00	0.10	1.0%
*R* _b,3_	1.067	0.014	0.58	5.67	54.5%
*m*_x,4_/mg	48.991	0.049	1.00	0.10	1.0%
*m*_y,4_/mg	197.813	0.198	1.00	0.10	1.0%
*m*_z,4_/mg	147.505	0.148	1.00	0.10	1.0%
*R* _b,4_	1.057	0.008	0.58	2.01	19.4%

**Table 6 foods-13-02377-t006:** Definition of the MCS for the MSA–IDMS with 4 blends including the binary blend 1. Masses of the PT 30110 sample (*FERA/FAPAS*^®^) *m*_x_, reference solutions *m*_z_ (*N* = 1) and the ^13^C_3_–MEL spike solution *m*_z_ (*N* = 1) are simulated for each standard addition blend 1–4. Sensitivity factors *c_i_* and the indices are calculated for experimental data 1/*R*_b*,i*_ (*N* = 2) according to NIST/SEMATECH (2012).

Measurand	Value	*u*	*c_i_*	*c_i_*^2^·*u*_r_^2^/10^−5^	Index
*m* _x1_	500.770	0.501	1.00	0.10	0.12%
*m* _y1_	99.340	0.099	1.00	0.10	0.12%
*m* _z1_	0.000	0.000	–	0.00	–
*R* _b,1_	10.428	0.209	0.71	20.00	24.60%
*m* _x2_	502.790	0.503	1.00	0.10	0.12%
*m* _y2_	101.460	0.101	1.00	0.10	0.12%
*m* _z2_	49.430	0.049	1.00	0.10	0.12%
*R* _b,2_	10.877	0.218	0.71	20.00	24.60%
*m* _x3_	500.760	0.501	1.00	0.10	0.12%
*m* _y3_	100.460	0.100	1.00	0.10	0.12%
*m* _z3_	99.320	0.099	1.00	0.10	0.12%
*R* _b,3_	11.787	0.236	0.71	20.00	24.60%
*m* _x4_	507.570	0.508	1.00	0.10	0.12%
*m* _y4_	100.670	0.101	1.00	0.10	0.12%
*m* _z4_	149.410	0.149	1.00	0.10	0.12%
*R* _b,4_	12.822	0.256	0.71	20.00	24.60%
*w* _z_	10.080	0.020	0.71	0.20	0.25%

**Table 7 foods-13-02377-t007:** Results of the MCS using the setting in Table 5 compared to the results of the GUM approach (GUM Workbench).

	MCS	GUM
*a* _0_	2.0534 ± 0.0355	2.0541 ± 0.0359
*a* _1_	1.6202 ± 0.2101	1.617 ± 0.211
Index (*R*_b,1_)	24.7%	41.0%
Index (*R*_b,2_)	24.7%	6.3%
Index (*R*_b,3_)	24.7%	3.8%
Index (*R*_b,4_)	24.7%	48.4%
*w*_X_/mg/kg (95% confidence, *k* = 2)	12.8 [+4.8, −2.9]	12.8 ± 3.7

## Data Availability

The original contributions presented in the study are included in the article, further inquiries can be directed to the corresponding author.

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
