# Peer review of "Combination of Standard Addition and Isotope Dilution Mass Spectrometry for the Accurate Determination of Melamine and Cyanuric Acid in Infant Formula"

_foods, 2024, doi:10.3390/foods13152377_

Round 1

Reviewer 1 Report

Comments and Suggestions for Authors

In the melamine scandal of the early 2000s, different companies in the dairy industry cheated their products by using chemicals to pretend that nitrogen levels were higher, and the deliberate adulteration of melamine (MEL) and cyanuric acid (CYA) not only resulted in non-negligible animal and human deaths, but also led to multiple food recalls and widespread public outrage. Today, both compounds are still used in large quantities as bulk chemicals, with MEL being used as raisins in the manufacture of laminates, plastics, glue, tableware, etc. CYA is a byproduct of the industrial use of melamine, and in outdoor swimming pools it is used as bleach, disinfectant or, most famously, chlorine stabilizer. Both of these compounds are chemicals with high nitrogen content, which can lead to misinterpretation of data from non-specific total protein measurement methods such as Kjeldahl nitrogen determination, so this study established a method for detecting melamine and cyanuric acid in infant formula based on the 2D-LC method, which can obtain more accurate results with better selectivity and sensitivity.

The following are comments on this paper:

1. The advantages and disadvantages of other methods for the detection of melamine (MEL) and cyanuric acid (CYA) and the improvement of this method are not introduced.

2. The international standard limits of melamine (MEL) and cyanuric acid (CYA) in infant milk powder are not introduced.

3. According to the description in the abstract, "a rapid screening method for melamine and cyanuric acid in infant formula milk powder was established", but the "rapid" feature was not reflected in the paper.

4. It is suggested that the optimization of 2D-LC method over 1D-LC method should be analyzed in detail in the conclusion.

5. Figure 3 is not clear, please provide a clear picture.

Comments on the Quality of English Language

Minor editing of English language required.

Author Response

Thank you very much for your valuable comments. We hope, we were able to further improve this manuscript with the help of your comments.

Comments 1: The advantages and disadvantages of other methods for the detection of melamine (MEL) and cyanuric acid (CYA) and the improvement of this method are not introduced.

Response 1: Revised as requested (please see also number 3).

Comments 2: The international standard limits of melamine (MEL) and cyanuric acid (CYA) in infant milk powder are not introduced.

Response 2: Revised as requested

Comments 3: According to the description in the abstract, "a rapid screening method for melamine and cyanuric acid in infant formula milk powder was established", but the "rapid" feature was not reflected in the paper.

Response 3: Revised as requested. The “rapid” feature lies on the one hand in the uncomplicated sample preparation and on the other hand in the 2D-LC technique, which makes it possible to separate melamine and cyanuric acid from matrix components in the 1st dimension of LC and to measure the desired components quantitatively without interferences on the 2nd dimension of LC.

Comments 4: It is suggested that the optimization of 2D-LC method over 1D-LC method should be analyzed in detail in the conclusion.

Response 4: Revised as requested

Comments 5: Figure 3 is not clear, please provide a clear picture.

Response 5: Revised as requested. However, in a review process, figures are often not in the highest resolution.

Comments 6: Comments on the Quality of English Language. Minor editing of English language required.

Response 6: Revised as requested. We gain gone through the text with the help of a native speaker.

Reviewer 2 Report

Comments and Suggestions for Authors

This paper combines an isotopically labelled internal standard and the standard addition method for the accurate determination of melamine and cyanuric acid in infant formula by LC-MS/MS. The added value of this paper is the metrological development (method validation and uncertainty calculation).

The paper is well written in English. However, it is rather difficult to follow. The objectives are clearly stated in the abstract and the introduction. The rest of the manuscript should be improved to facilitate understanding and further application.

Two minor corrections: TLC (page 2, line 57) should be defined. Page 4 (line 181) please correct "multiple reaction monitor" to "multiple reaction monitoring".

The figures should not appear in section 3.2. In my opinion, they should appear progressively as they are mentioned in the results and discussion.

The authors have prepared reference samples. Is the trueness of the method verified? I have seen a calculation of uncertainty. However, the assessment of traceability is not specified (with, for instance, recovery studies and t-test, confidence intervals and calculation of p-values)

Why is 2D LC used in the manuscript? I wonder if classical LC-MS/MS is not sufficient to calculate the concentration of both compounds. The compounds are correctly separated and MRM is used for quantification.

The method uses one-point calibration? However, Figure 1 shows the classical standard addition calibration model combined with isotope dilution. One-point calibration has been compared with the classical calibration model? Calibration models are usually preferred to ensure that the method is linear. In addition, linearity is not always assessed in LC combined with mass spectrometry. Sometimes, quadratic models are used for quantification.

Figure 4 shows the box plots of areas obtained from 1D and 2D chromatography. Are they obtained from precision studies? (Experimental variability of the ratios obtained when analysing several times the validation samples?). Perhaps an standard deviation of this variability could be given for the calculation of uncertainty.

Uncertainty is calculated using Monte Carlo simulations. I think that validation and uncertainty are the added value of the manuscript. In my opinion, it is quite difficult to understand how this calculation is done. It would also be interesting to calculate uncertainty following the error propagation law proposed by GUM and Eurachem. From Table 3 I conclude that the coefficients “ci” could probably be related to the calculation of partial derivatives and could be combined to give an overall standard uncertainty. However, all this is very difficult to see in the paper. A cause and effect diagram (fish and bond) could also be provided for each methodology. The contribution of each source to the uncertainty budget should also be more clearly identified and calculated. One-point calibration could also be compared with the uncertainty of classical calibration models.

Author Response

Comments 1: 

This paper combines an isotopically labelled internal standard and the standard addition method for the accurate determination of melamine and cyanuric acid in infant formula by LC-MS/MS. The added value of this paper is the metrological development (method validation and uncertainty calculation).

The paper is well written in English. However, it is rather difficult to follow. The objectives are clearly stated in the abstract and the introduction. The rest of the manuscript should be improved to facilitate understanding and further application.

Response 1: Thank you very much for your valuable comments. We hope, we were able to further improve this manuscript with the help of your comments.

Comments 2: Two minor corrections: TLC (page 2, line 57) should be defined. Page 4 (line 181) please correct "multiple reaction monitor" to "multiple reaction monitoring".

Response 2: -> Revised and corrected as advised.

Comments 3: The figures should not appear in section 3.2. In my opinion, they should appear progressively as they are mentioned in the results and discussion.

Response 3: -> Revised and corrected as advised.

Comments 4: The authors have prepared reference samples. Is the trueness of the method verified? I have seen a calculation of uncertainty. However, the assessment of traceability is not specified (with, for instance, recovery studies and t-test, confidence intervals and calculation of p-values)

Response 4: 

We used the pure CRM solutions of MEL/MEL-13C3 and CYA/CYA-13C3 for the standard addition method to check the validity eq. 8 and the relation between intercept and slope, which should be equal in case of matching contents of  sample and added standard solution (s. fig. 6). The results are in very good agreement to the expected values and confirm the traceability of the method itself to a given CRM, since all masses are determined by metrological process as well. t-Tests for MEL and CYA are added.  

Same is done for the matrix, which was added with the same CRM solutions. A comparison between pure solutions and „spiked“ matrix can be seen as „recovery“. We showed also results of the CRM PT 30110 to demonstrate the traceability to this material and another validated primary method (EMD-IDMS). A t-test is also added.

Confidence intervals (95% confidence) can be taken from the distributions of the Monte Carlo simulations (s. fig. 7 and 8).

Comments 5: Why is 2D LC used in the manuscript? I wonder if classical LC-MS/MS is not sufficient to calculate the concentration of both compounds. The compounds are correctly separated and MRM is used for quantification.

Response 5: Thanks for the comment. we have tried to better incorporate the advantages of 2D LC into the manuscript. The primary aim is to demonstrate a method in which two sought-after compounds can be quantitatively determined within a protein- and carbohydrate-rich matrix without a complicated sample preparation.

Comments 6: The method uses one-point calibration? However, Figure 1 shows the classical standard addition calibration model combined with isotope dilution. One-point calibration has been compared with the classical calibration model? Calibration models are usually preferred to ensure that the method is linear. In addition, linearity is not always assessed in LC combined with mass spectrometry. Sometimes, quadratic models are used for quantification.

Response 6: We compared the Standard Addition-IDMS method with the EMD-IDMS method, which is a single-point calibration of a matched sample and calibrator blend. As correctly mentioned, linearity must be demonstrated. Additionally, these calibration curves (fig. 2) are very useful for both EMD-IDMS and MSA-IDMS. The compositions of the calibrator mixture and the amounts for the standard addition can be derived from these curves and help to avoid several weighing cycles until matching is achieved. 

Comments 7: Figure 4 shows the box plots of areas obtained from 1D and 2D chromatography. Are they obtained from precision studies? (Experimental variability of the ratios obtained when analysing several times the validation samples?). Perhaps an standard deviation of this variability could be given for the calculation of uncertainty.

Response 7: Yes, the data for fig. 4 was obtained by repeatability experiments of the same sample solution. It represents the uncertainty of RB and RBc. This was added to the description. The actual uncertainties of these quantities can be found the results, e.g. tab. 2.

Comments 8: 

Uncertainty is calculated using Monte Carlo simulations. I think that validation and uncertainty are the added value of the manuscript. In my opinion, it is quite difficult to understand how this calculation is done. It would also be interesting to calculate uncertainty following the error propagation law proposed by GUM and Eurachem. From Table 3 I conclude that the coefficients “ci” could probably be related to the calculation of partial derivatives and could be combined to give an overall standard uncertainty. However, all this is very difficult to see in the paper. A cause and effect diagram (fish and bond) could also be provided for each methodology. The contribution of each source to the uncertainty budget should also be more clearly identified and calculated. One-point calibration could also be compared with the uncertainty of classical calibration models.

Response 8: 

Eurchem Guide “Quantifying Uncertainty in

Analytical Measurement”, 3rd ed, 2012 describes the way how we calculated the Monte Carlo simulation (MCS) in annex E. The calculation of the plot points according to eq. 6 in combination with the linear regression for eq. 8 is very complex and the simulation the only way to cover all contributions. The attachment of the calculation sheet may help to follow the calculation steps easier and could be used as template for others.
